# An Innovation Framework of Medical Organic Cannabis Traceability in Digital Supply Chain

Weerapat Pookkaman and Taweesak Samanchuen *

Technology of Information System Management Division, Faculty of Engineering, Mahidol University, Nakhon Pathom 73170, Thailand
* Correspondence: taweesak.sam@mahidol.ac.th

**Abstract:** Cannabis is increasingly accepted by medical organizations for medicinal and research purposes. A traceability system is required for monitoring and controlling the use of cannabis. This work aimed to investigate the relationship between critical success factors (CSFs) for creating the innovation framework that affects the implementation of the cannabis traceability system. These factors are identified based on the digital supply chain by structural modeling. Additionally, the issue of patients' safety is a crucial factor that needs to be considered. Total Interpretive Structural Modeling (TISM) and fuzzy MICMAC analysis techniques were applied to investigate the relationship between CSFs. Finally, the 14 CSFs affecting cannabis traceability systems were determined. The research results discovered that system reliability is the most significant key factor with the greatest influence and contribution to the achievement. The five CSFs based on TISM comprise harmonized strategic orientation, inter-organization collaboration, standardization, government regulations, and customer awareness. According to the conclusive dependence–driving power diagram, they have the most driving power in the implementation and operation traceability system. The innovation framework helps to establish the traceability system in a way that works well. This will build trust in the supply chain and prevent abuse and substandard products.

**Keywords:** traceability; medical cannabis; digital supply chain; TISM; fuzzy MICMAC

## 1. Introduction

Cannabis is a medicinal plant originating in Asia, and it is currently cultivated and used worldwide. It has become popular today due to its therapeutic properties and psychoactive substances. Cannabis used to be a part of Asian culture as an ingredient in many food recipes. Moreover, it has been used as traditional medicine; for instance, laborers used it as a muscle relaxant. Nevertheless, some people misuse cannabis for euphoria, extreme feelings of happiness, and excitement, leading to cannabis addiction, which has been increasing continuously. Finally, cannabis was banned worldwide in 1900 [1].

Cannabinoids extracted from cannabis have significant medical benefits, including the ability to treat several non-contagious diseases. Cannabidiol (CBD) is a non-toxic substance in medical cannabis, acting as an adjunct to the prescription to treat chronic illness. Recent research shows that the combination of CBD and opiates leads to signs of pain relief. In addition, it can decrease the use of opiates so their harmful effects can be avoided [2]. Additionally, cannabidiol has a potential treatment for chronic diseases such as diabetes mellitus, cancer, and Huntington's disease. It is also used to treat neurological disorders such as Parkinson's and Alzheimer's disease [3,4]. Despite the potential benefits of cannabis, it has undesirable effects on blood pressure, mood disorders, and addiction. Therefore, practitioners should weigh the benefits of medical cannabis against its risks.

The national laws and regulations of cannabis differ from country to country. Most of them are still unclear on monitoring data and control plans. For example, federal and state legislators enacted numerous rules in the United States that differed widely from one

another. Some states allow cannabis for medicinal purposes only. Therefore, cannabis can be grown and produced in those states. However, this affects the tracking and tracing system and cannot be easily implemented in the nationwide supply chain. Furthermore, laws and regulations hinder the practical approach to addressing consumer dissatisfaction and accessing cannabis products when needed [5]. Each country should consider improving the cannabis laws and regulations for medical benefits, such as establishing medical cannabis traceability standards.

The medical cannabis production processes must control the quality of the production system from upstream to downstream, following the government regulator [6]. Good Manufacturing Practice (GMP) is the standard to ensure product quality and consistency. The World Health Organization (WHO) has recommended that the herbal industry complies with the GMP. The standard ensures that the production system reaches pharmaceutical quality and minimizes production risks. For example, cross-contamination with pathogens, chemicals, pesticides, and physical contaminants will not occur, and mix-ups risk causing unwanted side effects on humans. In addition, the traceability standard, according to which the company must ensure the process by tracing the material or product at least one step backward and one step forward, is necessary [6,7].

The traceability system is one of the excellent solutions to ensure cannabis medicine quality and proper use (not abuse or misuse). As mentioned earlier, a traceability system for the organic medical supply chain can build confidence in the patients and medical standards. Additionally, to create a seamless communication platform among various stakeholders, the digital supply chain (DSC) was used to develop the conceptual framework. Many successful cases of organic products indicate added value and build customer trust. However, these products require excellent logistics management, connecting the upstream and downstream stakeholders. Furthermore, this work prioritizes patient safety as an essential factor and is added to prevent risks, errors, and harmful effects. The pharmacovigilance guidelines for seven herbal medicines of the WHO were applied to this study [8,9].

The proposed innovation framework will be fundamental for further implementing a MOCT system. The proposed framework shows not only the relationship between the factors but also the driving power of each factor. Therefore, we can understand the relationship between these factors and implement the MOCT system properly. In a globalized society, customers are aware of the quality and standard of products and must be able to check information in real-time. This work will help ensure that the traceability system's planning and implementation are complete and sustainable.

While theoretical foundations for medical organic cannabis traceability (MOCT) in the DSC were developed, three questions were posed: what are CSFs, how are factors related, and what type of dependence and driving power do CSFs have? Therefore, this work has three main objectives:

(1)    To identify the CSFs of MOCT in the digital supply chain;
(2)    To use the innovation framework of CSFs of MOCT in the DSC;
(3)    To classify the CSFs based on their driving and dependence powers.

The remaining sections of this paper are organized as follows. Section 2 presents a literature review of the conceptual framework and explores factors that lead to the success of the traceability system. Section 3 describes the research process, and Section 4 explains the confirmatory factors and model design with TISM analysis and fuzzy MICMAC analysis. Section 5 discusses the critical results from the TISM model and fuzzy MICMAC diagram. Finally, the conclusions, which includes limits and future research, are presented in Section 6. The results will greatly benefit the medical organic cannabis supply chain by building trust with stakeholders and patients alike and preventing misuse through a traceability system.

## 2. Literature Review

Related theories and works were reviewed to develop the proposed framework. As a result, this section is divided into two subsections. The first subsection addresses the

supply chain theory, which describes the background of the conceptual framework. In addition, the stakeholders related to MOCT are defined in this subsection. The second subsection presents a review of the literature on CSFs of MOCT. The review was conducted using a systematic approach based on the technology–organization–environment (TOE) theory. The details can be found in Section 2.2.

### 2.1. Supply Chain

The review started from the dimension of supply chain analysis, in which the stakeholders participating in producing organic medical cannabis from upstream to the end-users were considered. The related stakeholders include planting, harvesting, warehouse, production processing, distribution, dispensing, and patients.

Moreover, the regulations, standards, and technological dimensions were included to investigate the related factors. Pharmacovigilance, which is concerned with detecting, assessing, understanding, and preventing adverse drug effects from medical cannabis, was taken into account [8,9]. Therefore, pharmacovigilance is essential for the complete presentation of the medical supply chain concept. The DSC was created to support real-time information [10] with the consideration of inter-organizational collaboration [11]. When the idea is complete and accurate, the research constructs the conceptual framework of MOCT in the DSC for research development, as shown in Figure 1. The inner circle represents the players/stakeholders in the medical organic cannabis supply chain. Connecting lines with arrows illustrate the delivery of products and services, and dotted lines represent real-time information exchange between stakeholders using cloud computing. The outer frame represents the rules, regulations, and standards required to operate the traceability system. The work considers studying organic cannabis because of patients' expectations. It requires natural therapy without chemical contamination. In addition, choosing organic cannabis gives patients the confidence that the products conform to organic standards (free from pesticides, fungicides, or heavy metals) [12,13].

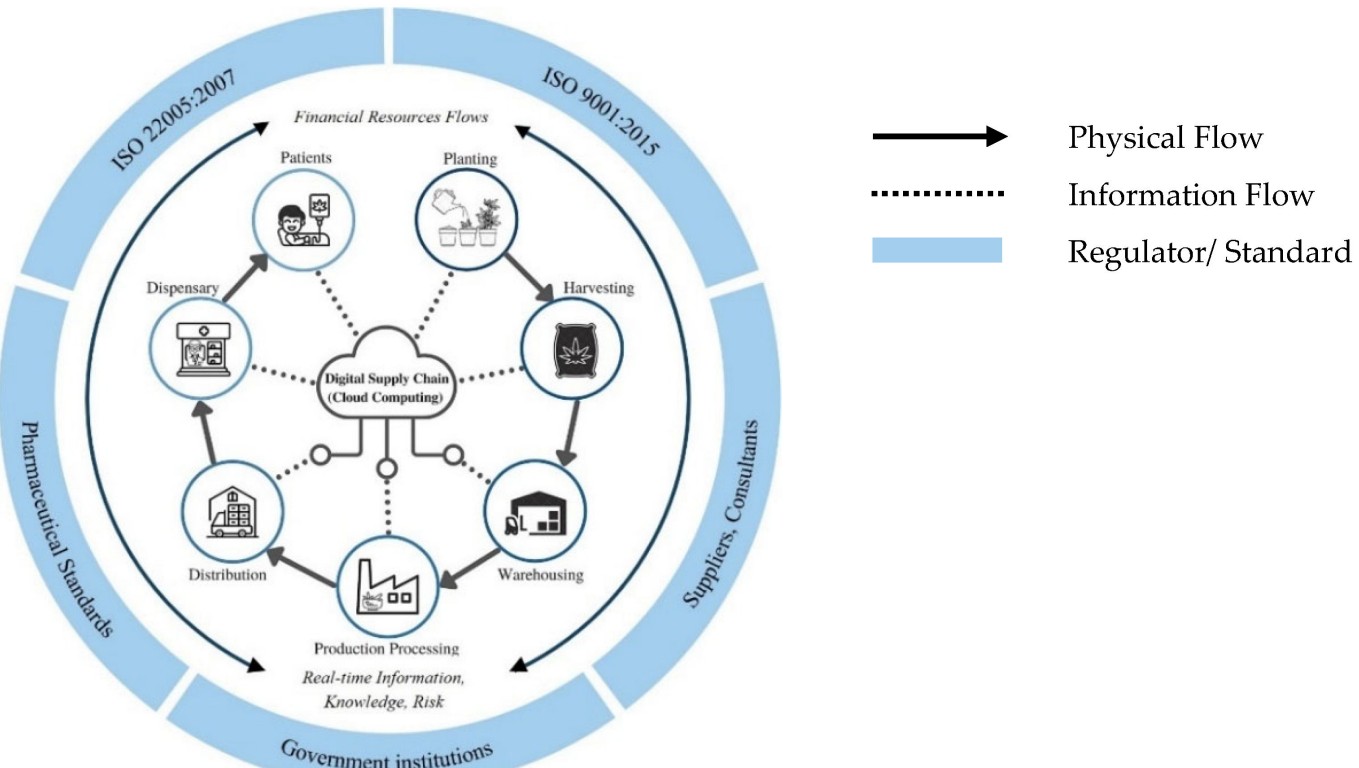

**Figure 1.** Conceptual framework of MOCT in the DSC.

Currently, the market is complex and fluctuates due to dynamics and increasing technological advances. The DSC strategy is one of the solutions that increase competitive advantage. It encourages the stakeholders in the supply chain to exchange information and also supports them in adjusting the production plan in real time [14]. The resulting efficiency leads to a reduction in production costs, storage costs, and an increase in the response information for increasing flexibility in supply chain management. Examples of organizations adapted to digitalization in the supply chain are Alibaba, Amazon, BMW, Lufthansa, and DHL [15–17].

The traditional supply chains transfer the products and services from the upstream actor to the last player through the stakeholders in the following supply chain. On the other hand, the information flow of demand will be transferred from customers to upstream players. It can be concluded that the traditional supply chain was delayed in transmitting information and was not flexible. Therefore, when customers have to adjust the quantity or cancel the order, the manufacturer cannot change the manufacturing plan in real time. By contrast, the DSC links all stakeholders in the supply chain for demand information and other information in one database in real time via cloud computing with high security such as blockchain [10,18]. It ensures that players trust the data's safety and security for the exchange. As a result, the company can adapt production planning to changing information in time.

### 2.2. Review of Critical Success Factor

This section aims to determine the factors that affect the MOCT in the DSC to create an innovation framework. The framework will be helpful for interested organizations in preparing resources and layout strategies for the whole supply chain before putting in place a traceability system. The extraction and grouping of factors in this work using the TOE theory was created by Tornatzky and Fleisher [19]. It describes the characteristics and how the organizational context influences the implementation of technology and its potential. The TOE framework is an organization-level theory demonstrating how three different elements of an organization's context influence adoption judgments. Therefore, we used this conceptual framework to reduce duplication and group the factors that affect the use of the traceability system. This framework consisted of three elements: the technological context, the organizational context, and the environmental context. These contexts were posited to influence the implementation of traceability technology [20]. The systematic literature review method was applied to determine the CSFs that affect the MOCT system. The method was separated into categories, that is, pharmaceutical traceability, pharmacovigilance traceability, food traceability, organic traceability, and the DSC, to search for factors affecting the success of traceability implementation.

Multiple keyword factors related to CSFs of MOCT in the DSC were hypothesized to exist, and a systematic literature review was managed by searching for "traceability + supply chain + medicine + drug + factors"; "traceability + pharma* + factors"; "pharmacovigilance + policy + implementation"; "pharmacovigilance + success + factors"; "traceability + food + factors"; "traceability + food + supply chain"; "organic + traceability + factors"; "organic + traceability + supply chain"; "digital supply chain + implementation + factor"; "implementing systems + digital supply chain + factors". We focused on the factors affecting the traceability system in trustworthy databases, including Scopus, Web of Science, books, and organization reports. Due to limited information on medical traceability, the food traceability factor was instead considered. Fourteen factors are shown in Table 1, where the scope of literature is categorized into five groups: pharmaceutical traceability, pharmacovigilance traceability, food traceability, organic traceability, and digital supply chain. At the top of the table are factors that were classified into three groups according to the TOE theory. After a literature review, this research proposed an innovation framework for MOCT that includes all critical success factors as shown at the end of Table 1. In Table 2, the definitions of CSFs are given with references.

**Table 1.** Empirical evidence for identifying factors.

| Reference | Technology | | | Organization | | | | | | | Environment | | | |
|---|---|---|---|---|---|---|---|---|---|---|---|---|---|---|
| | Supply Chain Visibility | System Reliability | Technology Maturity | Harmonized Strategic Orientation | Financial Capabilities | Staff Awareness | Manager Attention | Communication | Inter-Organizational Collaboration | Training | Standardization | Government Regulation | Government Support | Customer Awareness |
| **Pharmaceutical Traceability** | | | | | | | | | | | | | | |
| [21] | √ | | √ | | √ | | √ | √ | √ | | √ | √ | | |
| [22] | √ | √ | √ | | √ | | √ | √ | √ | | √ | √ | √ | √ |
| [11] | | | √ | √ | | | | | √ | √ | √ | | √ | √ |
| [23] | | | √ | | | √ | | | | | √ | | | |
| [24] | | √ | √ | | | | | | | | √ | √ | | |
| [25] | | √ | | | | | | | | | √ | | | |
| **Pharmacovigilance Traceability** | | | | | | | | | | | | | | |
| [26] | | | | √ | | | | √ | | | √ | | | |
| [27] | | | √ | | √ | | | | | √ | | √ | √ | √ |
| [28] | | | | | | | | | √ | | √ | √ | √ | |
| [29] | | | | | | √ | | √ | | | | √ | | |
| [30] | | | | | | √ | | √ | | √ | | | | √ |
| [9] | √ | | | | | | | | | | | | | |
| **Food Traceability** | | | | | | | | | | | | | | |
| [31] | | | √ | √ | √ | √ | √ | | | √ | √ | | | |
| [32] | | | √ | | | | √ | | | | √ | | √ | |
| [33] | | | | | | | | | | | | | | √ |
| [34] | | | | √ | | | | | | | | √ | | |
| [35] | | | √ | | | | √ | √ | √ | √ | √ | | √ | |
| [36] | | √ | √ | √ | | | | √ | | √ | √ | √ | √ | √ |
| [37] | | √ | √ | | | | | | | √ | | | | |
| [38] | | | √ | √ | | | | | | | | √ | | √ |
| [39] | √ | | | | | | | | | | | | | |
| [40] | | | | | | | | | | | | √ | | √ |
| **Organic Traceability** | | | | | | | | | | | | | | |
| [41] | √ | √ | | | | √ | √ | | | √ | √ | √ | √ | |
| [42] | | | | | | √ | | √ | | √ | √ | | | |
| [43] | | | | | | | | | | | | | | √ |
| [44] | √ | | | √ | | | | | √ | | √ | √ | | √ |

**Table 1.** *Cont.*

| Reference | Technology | | | Organization | | | | | | | | Environment | | |
|---|---|---|---|---|---|---|---|---|---|---|---|---|---|---|
| | Supply Chain Visibility | System Reliability | Technology Maturity | Harmonized Strategic Orientation | Financial Capabilities | Staff Awareness | Manager Attention | Communication | Inter-Organizational Collaboration | Training | Standardization | Government Regulation | Government Support | Customer Awareness |
| Digital Supply Chain | | | | | | | | | | | | | | |
| [15] | √ | | √ | √ | | √ | | √ | √ | | √ | | | |
| [45] | √ | | | √ | | √ | √ | | √ | | | | | |
| [10] | √ | | √ | √ | | | | | | | √ | | | √ |
| [46] | √ | | | √ | √ | | √ | √ | √ | | | | | |
| [16] | √ | | √ | | | | | √ | √ | | | | | |
| [17] | √ | | √ | √ | | | √ | √ | √ | | | | | |
| [47] | | √ | | √ | | | √ | √ | √ | √ | √ | | | |
| Proposed Innovation Framework for MOCT | √ | √ | √ | √ | √ | √ | √ | √ | √ | √ | √ | √ | √ | √ |

### 2.2.1. Supply Chain Visibility

Supply Chain Visibility allows all parties in the supply chain to immediately visualize all information, consisting of the number of inventories, production plans, standards used, changes in demand, and other information simultaneously. It enables the management and operational process industries to reduce the bullwhip effect and increase process optimization in the supply chain [22]. Therefore, the DSC is a mechanism that supports more rapid and smooth operation along the value chain [45,46]. It is suitable for the pharmaceutical industry, in which the products expire quickly, and large stocks of the products are not convenient [46]. DSC is consistent with the process of pharmacovigilance as the visibility of the process allows all related parties to know critical information through communication channels or appropriate technology channels immediately, resulting in patient safety [9]. In summary, the increasing visibility of information across the supply chain facilitates the stakeholders' ability to be more confident in the information. They can check product details and quantity status in real time, resulting in effective decision-making.

### 2.2.2. System Reliability

System Reliability refers to the reliability of information and technology devices used in traceability systems to track and trace electronic input devices. A traceability system is one that connects all activities and actors in the supply chain. If actors upstream record incorrect data, it will also result in erroneous downstream data [24,25,36]. There are many electronic input devices and technologies in which different technologies may cause incompatibility problems among companies in the supply chain, such as causing intermittent interruptions of data transmission, resulting in incomplete information [37,41]. Normally, users of the traceability system are concerned about data quality. It results from a misunderstanding or inaccuracy of information from unfamiliar technology [22]. In short, System Reliability is an important part of setting up a traceability system because if the system provides wrong vital information, it will hurt the organization's credibility and put patients at risk.

### 2.2.3. Technology Maturity

Implementing a traceability system using a state-of-the-art system will increase work efficiency in the long term. The research suggests using blockchain technology to ensure data integrity and increase stakeholder confidence throughout the supply chain [21] or using smartphone applications instead of barcode reader devices to save the budget for purchasing devices [23]. In addition, the maturity of technology will increase the efficiency of analyzing big and complex data for fast analysis, reliable data, and information accuracy [10]. If technological devices are accessible and widespread, customers will participate in product inspections themselves, affecting product satisfaction and creating customer retention [24]. Finally, the selected technologies used in the traceability system should be suitable for the product, the organization, and the supply chain, resulting in a positive effect on performance [35]. Technology maturity will help the company facilitate the gathering of information. It can also quickly analyze complex data and support protection data all along the supply chain, which gives customers more confidence in the product.

### 2.2.4. Harmonized Strategic Orientation

Strategic orientation consists of priorities, goals, strategies, and vision. It ensures that its activities and others are aligned with the organization. All organizations that must operate together should have predefined objectives or project strategies [47] to visualize the guidelines for collaboration [11]. The strategic objectives should be designed to align with the customer's needs and the ability of corporate executives [10,47]. Before starting the project, each organization should agree to align and clarify roles and responsibilities [15,31,45]. In addition, the pharmaceutical industry's efforts should also be restructured by defining policies to implement in the DSC [17]. It is a plan to drive the organization to designate the direction of its business. Actions should be designed so that all stakeholders are aware of them and understand them in order that all parties have the same goals and the same direction of operation across the supply chain.

### 2.2.5. Financial Capability

There are three dimensions to be considered for financial capability based on the TOE framework. The first dimension consists of the organization's cost sections, such as wages, coordination, and training costs. The second includes the technology sectors, such as traceability software and devices, and the third includes the environmental sectors, consisting of consulting fees and setting standards [21,32]. Furthermore, maintenance of the traceability system is another essential cost for all companies [41]. As mentioned, the company should continuously and sufficiently support the money invested in obtaining an efficient and sustainable system [36,46]. A traceability system may reduce adverse events in pharmacovigilance, thereby saving the budget for managing patient problems [39]. In summary, the company's Financial Capability is the key factor driving the organization. It is needed to support operations for investment in human resources, technology, and regulations for efficiency, continuity, and sustainability.

**Table 2.** Critical success factors (CSFs) for MOCT.

| No | Factors | Definition | References |
|---|---|---|---|
| 1. | Supply Chain Visibility (C1) | Ability to track raw materials and customer's demands in the supply chain in real time from upstream to downstream | [9,10,15–17,21,22,39,41,44–46] |
| 2. | System Reliability (C2) | The efficiency and quality of being trusted that a traceability system will accurately perform specified tasks under the stated environmental conditions | [22,24,25,36,41,47] |
| 3. | Technology Maturity (C3) | Mature technology development to improve the efficiency of business performance in the digital supply chain | [10,11,15–17,21–24,27,31,35,38] |
| 4. | Harmonized Strategic Orientation (C4) | The harmonized plan that ensures that the organizations in the supply chain pursue the same future business goals | [11,15,17,26,31,34,36–38,44–47] |
| 5. | Financial Capability (C5) | The state of being capable or power of finance of an organization | [21,22,27,31,32,36,41,46] |
| 6. | Staff Awareness (C6) | The staff understands that the processes of work and information recording in the traceability system are essential according to the policy, goals, and vision of the organization | [15,23,29–31,41,42,45] |
| 7. | Manager Attention (C7) | Persons responsible for managing or directing an organization and controlling the results of the job are key performance indicators | [17,21,22,31,32,35,42,45–47] |
| 8. | Communication (C8) | A process of exchanging information or ideas between people through a standard system of behavior or symbols and signs | [15–17,21,22,26,29,30,35,36,46,47] |
| 9. | Inter-organizational Collaboration (C9) | Cooperation among business organizations in the same supply chain in a friendly and trusted manner | [11,15–17,21,22,35,44–47] |
| 10. | Training (C10) | The process of learning the skills through introducing, clarifying, and understanding a particular job or activity | [11,27,30,31,35–37,41,42,47] |
| 11. | Standardization (C11) | The process of ensuring that traceability systems of the same type all have the same basic features or technology type to assure consistency and regularity throughout the supply chain | [10,11,15,21,22,24–26,31,32,35,36,41,42,44,47] |
| 12. | Government Regulations (C12) | The rules of government to control and make sure that business is operating according to a standard or requirements of operation | [21–24,27,29,34,36,38,40,41,44] |
| 13. | Government Support (C13) | The ability of the government to help or encourage the organization to implement a traceability system to succeed | [11,22,27,32,35,36,41] |
| 14. | Customer Awareness (C14) | The realization of a customer of the information about products, ingredients, services, and customer rights, which will support making the right decision and making the right choice for safety | [10,11,22,27,30,33,36,38,40,43,44] |

### 2.2.6. Staff Awareness

With awareness and discipline, staff at an operational level will drive the traceability implementation system successfully. However, the company should inform staff of the benefits [15,41] and implement training programs to raise awareness. According to the research, if staff had personal perceptions about traceability, the majority (more than 50%) would respond to the matter [42]. In actual operations, the organization sometimes employs staff from outsourcing companies where workers are often replaced, causing a lack of awareness. As a result, the data recording in the traceability system is incomplete or inaccurate [31]. Consistent with the hospital pharmacy, a lack of awareness about its operations led to a pharmacovigilance report failure [23]. Research indicates that educating during the study period about raising awareness is one of the solutions that help make staff more aware [30]. In conclusion, if the company can raise awareness for staff at work, it will help increase work efficiency and ensure the successful implementation of the traceability system.

### 2.2.7. Manager Attention

The managers are responsible for directing, controlling, and monitoring the staff or employees in following the plan and timeline of the company. Good managers have to lead the team in the organization to achieve the business objectives and be successful [22,45]. To successfully implement a traceability system in the supply chain, each organization should have experienced managers who can plan, exchange, and understand the plan before it is implemented [21,35,46]. The research found that most high-end user managers use the traceability system to improve product quality and maintain the quality of work processes [31]. As previously mentioned, top management must support money, facilities, and time sufficiently and continuously [47]. In summary, the manager is the person who is responsible for controlling and monitoring the process of the organization, and leading the team to accomplish the business plan and objectives.

### 2.2.8. Communication

Many organizations have discovered that a lack of Communication causes problems with exchanging data, such as incorrect demand and inventory data, resulting in an error in planning orders and delayed product delivery [21]. In the globalization era, communication channels facilitate employees' communication and the exchange of necessary information. Good communication results will improve work efficiency and customer satisfaction [46]. In addition, some manufacturers expect to communicate important information to customers through a traceability system [44]. Research suggests that performance indicators of action on pharmacovigilance identify practical measurements for communicating and exchanging stakeholder information [29]. Medical communication is essential, especially to provide timely advice to physicians, patients, and regulators, which reduces the risk of adverse drug reactions [30]. In conclusion, communication and communication frequency are crucial for working towards achieving objectives.

### 2.2.9. Inter-Organizational Collaboration

For traceability system implementation to succeed throughout the supply chain, it requires efficient and effective collaboration among the relevant actors [36]. The collaborative foundation in the supply chain is that the exchange of useful information and knowledge is required by each organization [15]. For the prerequisites of Inter-organizational Collaboration, the organizations have to trust each other and seek a partner with the knowledge, ability, and sufficient resources that are not much different from the company's own [11,35]. In exchange for information between organizations, signing the agreement with the exchanged information will be treated as confidential [21]. For example, exchanging information, sales forecasts, and production data through the DSC platform supports the data connection in real time between organizations through various channels [16,48]. This creates a competitive advantage for other supply chains in the disrupted business

world [22]. In conclusion, collaboration between actors from upstream to downstream will be strengthened and made competitively advantageous by trust between organizations.

### 2.2.10. Training

If the staff are trained, the operation will be performed correctly, smoothly, and quickly [11,36]. The research found that Training would ensure that staff are willing to obey and develop self-discipline [41]. Moreover, the study found that adequate quality management and food safety training successfully implement a traceability system [42]. Training may use internal personnel through several channels, such as on-site conferences [31,47]. After Training, organizations should build an awareness culture by following up continuously through social networks and instant-messaging applications [30]. The efficient training program results in not only skill development and awareness in personnel but also developing the efficiency of an implementation traceability system [35]. Effective Training will increase job productivity and assist the organization in deploying a traceability system successfully.

### 2.2.11. Standardization

Standardization is a tool that reduces the hurdle of interoperability in implementing a traceability system [21]. Before selecting vendors, partners who want to implement a traceability system in the supply chain must agree on common standards [47]. This is because the data come from several companies in the global supply chain with different organizational contexts, standards, and databases. Therefore, it is necessary to define common standards by setting data standards for methods, processes, and technology to a uniform standard so that all stakeholders can transmit and share data [24,36]. The research suggested that one of the crucial factors for adopting a successful traceability system was creating unified standards in the supply chain [32]. In each organization, every process must be inspected for quality, from raw materials to end-users [25,26,35]. To summarize, standardization is essential to ensure that everyone in the supply chain works correctly and consistently.

### 2.2.12. Government Regulations

The government should drive organizations to implement traceability systems through legislation and regulation [40]. Many countries' governments have devised implementing the traceability system as a tool for quality and safety [22,41]. Firstly, the supply chain actor should study the regulations that involve products, details of data to be stored, and data reporting [34]. In addition, international trade should consider harmonized international regulations to ensure the seamless operation of the traceability system in both domestic and global supply chains [36,38]. Research shows that the government's power to implement a traceability system made a greater contribution to the development of an effective traceability system than voluntary traceability systems [49]. Government Regulations have played a role in driving traceability systems because companies have to follow the food and drug industry's safety standards, which follow the laws and regulations set by the trade partner or international regulator.

### 2.2.13. Government Support

Governments should provide guidelines and policies that support the structure of traceability systems for organizations [11]. Each country may apply guidelines on implementing a traceability system from its food and drug administration inter-organization to be consistent throughout the global supply chain [32]. Furthermore, governmental organizations, such as the FDA, WHO, or IMPACT, may help the global supply chain by providing knowledge, certified documents, funding, equipment, training, technology, and tax concessions [35]. The research suggested that a traceability system was also a limitation for small and medium enterprises (SMEs) because funding, knowledge, and incentives from unrealized benefits were lacking [36]. Therefore, the government should support and

create guidelines for organizations to meet the standards in traceability. It also supports technology training and funding sources and facilitates administrative procedures.

### 2.2.14. Customer Awareness

Customers have become increasingly aware of the safety and integrity of food and pharmaceutical products due to emerging diseases [41]. Moreover, as a result of globalization, the supply chain develops into a DSC where customers want to know more about the safety of products, such as where the raw materials are from and what processes they have been through [11,33]. As a result, some customers are willing to pay more in exchange for reliable food. For features that cannot be checked visually, the traceability system allows customers to verify product information from the beginning to the end of the process [36,44]. Traceability systems create confidence in the customer due to the ability to retrieve information from the system [35]. Customers are increasingly aware of the products that they consume due to the perception of information. As a result, companies have to gather information to support a traceability system.

Based on the literature review, we found fourteen variables affecting MOCT in DSC. In the following process, experts confirmed whether these variables were CSF or not. Then, the CSFs were examined to establish a relationship between them by using TISM analysis.

### 3. Research Process

In this section, the research process of this work is described. Qualitative and quantitative methods were used. The research began with the factors' extraction process, then the factors' confirmation process, and finally the CSFs' relationship process, which are all presented in Figure 2. Firstly, the literature review extracted the factors and grouped them by using the TOE framework method. This method is an organization-level theory that demonstrates how three different elements of an organization's context influence adoption judgments. It consists of three elements: the technological context, the organizational context, and the environmental context. All three are posited to influence technical implementation. Multiple keywords related to MOCT in the DSC were defined for search clarity. A systematic literature review was performed by searching publications using "traceability", "pharmacovigilance", "DSC", "organic products", and related words. Then, the factors confirmed by academicians and industry experts were selected using a specific random sampling method from three groups of stakeholders, including three experts in organic cannabis cultivation, four experts in the control quality/regulator department, and three experts from the responsible department for dispensing cannabis to patients. All of the experts had a solid background in medical cannabis, as shown in Table 3.

This work adopted the Item-Content Validity Index (I-CVI) to confirm the level of agreement of factors. It is a good research tool for confirming social-science, pharmacy-administration, and health-science factors. After confirming these factors, they are called CSFs, affecting the success of implementing a MOCT system. TISM analysis is a systematic evaluation method for complex research. It establishes contextual relationships between factors based on the consensus expert's opinions. The dependence and driving power are relationships between CSFs to establish a TISM, which is upgraded from the ISM to explain the reason for the relationship better. Moreover, a fuzzy MICMAC analysis was used to identify deep relationships on a linguistic scale [38].

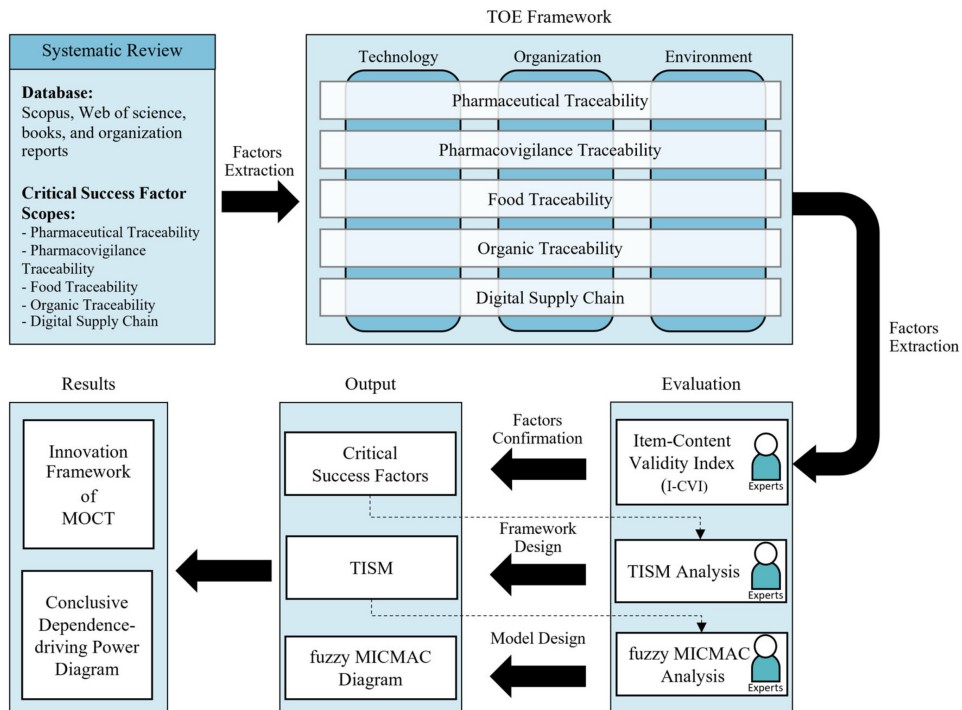

**Figure 2.** Schematic diagram of the research process.

**Table 3.** Experts' background in order from upstream to downstream expert.

| Experts | Expert Profile/Position | Area Expertise | Experience |
|---|---|---|---|
| Expert 1 | Academician, Researcher | Agriculture, Farming | >24 Years |
| Expert 2 | Organic and Cannabis Farmer, Consultant | Organic Cannabis | >6 Years |
| Expert 3 | Cannabis Consultant and Researcher | Cannabis Supply Chain | >5 Years |
| Expert 4 | Associate Professor, Cannabis Researcher | Logistics and Supply Chain | >20 Years |
| Expert 5 | Pharmacist, Government Pharmaceutical Organization | Cannabis System | >20 Years |
| Expert 6 | Pharmacist, Director of Division, Ministry of Health | Herbal Medicine | >19 Years |
| Expert 7 | Academician, Pharmacist, Consultant | Enterprise Architecture, Health System | >12 Years |
| Expert 8 | Academician, Pharmacist, Researcher | Thai Herbal Medicine | >30 Years |
| Expert 9 | Pharmacist, Professional Level | Herb Products for Patients | >25 Years |
| Expert 10 | Physician, Regional Manager | Oncologist | >12 Years |

## 4. Evaluation and Results

### 4.1. Agreement Index

Experts assessed the context and content validity of medical organic cannabis traceability factors. Table 4 presents the results of the experts' validity test, where E represents the order of the experts and C represents the sequence of factors. I-CVI calculation to find the relationship and agreement by item was divided into four items on a Likert scale, consisting of 4 = extremely relevant, 3 = quite relevant, 2 = somewhat relevant, and 1 = not relevant. For the test, the experts assessed each item based on personal opinion and experience. The accepted I-CVI value should not exceed 0.78 and will pass the criteria [50]. The following equation can be used to compute I-CVI:

$$I - CVI = \frac{A}{N},$$

(1)

where A is the number of experts in agreement.

**Table 4.** Agreement index calculation of CSFs.

| Factors | E 1 | E 2 | E 3 | E 4 | E 5 | E 6 | E 7 | E 8 | E 9 | E 10 | Agreement Number | I-CVI | k* | Evaluation |
|---------|-----|-----|-----|-----|-----|-----|-----|-----|-----|------|------------------|-------|------|------------|
| C1  | 4 | 4 | 4 | 4 | 3 | 4 | 4 | 3 | 4 | 4 | 10 | 1.00 | 1.00 | Excellent |
| C2  | 4 | 4 | 4 | 4 | 4 | 4 | 4 | 4 | 4 | 4 | 10 | 1.00 | 1.00 | Excellent |
| C3  | 4 | 4 | 4 | 4 | 2 | 3 | 4 | 2 | 4 | 4 | 8  | 0.80 | 0.79 | Excellent |
| C4  | 4 | 4 | 4 | 4 | 4 | 4 | 4 | 4 | 4 | 4 | 10 | 1.00 | 1.00 | Excellent |
| C5  | 2 | 4 | 2 | 4 | 3 | 3 | 4 | 3 | 3 | 4 | 8  | 0.80 | 0.79 | Excellent |
| C6  | 4 | 4 | 3 | 4 | 4 | 4 | 3 | 4 | 4 | 3 | 10 | 1.00 | 1.00 | Excellent |
| C7  | 4 | 4 | 3 | 4 | 4 | 4 | 4 | 4 | 4 | 4 | 10 | 1.00 | 1.00 | Excellent |
| C8  | 2 | 3 | 4 | 4 | 4 | 3 | 4 | 4 | 2 | 3 | 8  | 0.80 | 0.79 | Excellent |
| C9  | 4 | 4 | 3 | 4 | 2 | 3 | 4 | 2 | 4 | 4 | 8  | 0.80 | 0.79 | Excellent |
| C10 | 3 | 4 | 4 | 4 | 4 | 4 | 4 | 4 | 3 | 3 | 10 | 1.00 | 1.00 | Excellent |
| C11 | 4 | 4 | 4 | 4 | 3 | 4 | 3 | 3 | 4 | 4 | 10 | 1.00 | 1.00 | Excellent |
| C12 | 4 | 4 | 4 | 4 | 4 | 4 | 2 | 4 | 4 | 3 | 9  | 0.90 | 0.90 | Excellent |
| C13 | 3 | 4 | 2 | 4 | 4 | 3 | 4 | 4 | 3 | 2 | 8  | 0.80 | 0.79 | Excellent |
| C14 | 4 | 4 | 4 | 4 | 2 | 3 | 4 | 3 | 4 | 2 | 8  | 0.80 | 0.79 | Excellent |

The following is the formula and criterion: the value of Kappa (k*) is a Kappa statistic as supported by multi-rater Kappa statistics because it has a higher chance of being accepted as the validity factor by experts [51]. It can be interpreted as excellent (k* > 0.74), good (k* = 0.60–0.74), or fair (k* = 0.40–0.59) [50,52]. The value of Kappa (k*) has the following formulas:

$$K^* = \frac{(I - CVI - P_c)}{(1 - P_c)},$$ (2)

and

$$P_c = \left[\frac{N!}{A!(N - A)!}\right] 0.5^N,$$ (3)

where N is the number of experts [53].

This result shows that 14 factors (C1–C14) were excellent, which would be considered CSFs influencing the implementation of the MOCT system, as presented in Table 4. The accepted CSFs were used to explore the relationships in the following processes.

*4.2. Total Interpretive Structural Modeling Analysis*

TISM analysis is a technique developed to explain the relationship between complex systems in a way that is easier to understand and more convenient, such as supply chain management, change management, and performance management [54,55]. The TISM analysis comprises six steps to show how CSFs relate to each other so that an innovation framework can be made.

4.2.1. Development of Structural Self-Interaction Matrix (SSIM)

SSIM is the most important process and must be followed carefully to determine the relationship between CSFs. Experienced experts must consider the direct relationship between factors regardless of the latent relationship to eliminate the redundancy in the transitivity relationship. Explaining the principles of TISM processes can help reduce relationship errors. After the CSFs were confirmed, the pairwise relationship between the factors was performed using symbols to represent the relationship as follows [56]:

V: the relationship or influence "Ci" leads to "Cj",
A: the relationship or influence "Cj" leads to "Ci",
X: the relationship or influence "Ci" and "Cj" both lead to each other,
O: no relationship or influence between "Ci" and "Cj",

where Ci and Cj are the ith and jth CSFs, respectively.

All of these factors are shown in Table 5. The data were gathered from ten expert opinions, and the Delphi method was used to demonstrate the relationship between MOCT factors in the DSC.

**Table 5.** Structural Self-Interaction Matrix (SSIM).

| CSFs | C14 | C13 | C12 | C11 | C10 | C9 | C8 | C7 | C6 | C5 | C4 | C3 | C2 | C1 |
|------|-----|-----|-----|-----|-----|----|----|----|----|----|----|----|----|----|
| C1 | A | O | O | O | O | A | O | O | O | O | A | A | V | |
| C2 | O | O | O | O | O | O | A | A | A | O | O | A | | |
| C3 | O | A | O | O | V | A | O | O | O | A | A | | | |
| C4 | A | O | A | A | O | V | V | V | O | V | | | | |
| C5 | O | O | O | O | O | O | O | O | O | | | | | |
| C6 | O | O | O | O | X | O | O | X | | | | | | |
| C7 | O | O | O | O | V | O | O | | | | | | | |
| C8 | O | O | O | O | A | O | | | | | | | | |
| C9 | X | V | X | V | O | | | | | | | | | |
| C10 | O | O | O | O | | | | | | | | | | |
| C11 | O | O | A | | | | | | | | | | | |
| C12 | A | V | | | | | | | | | | | | |
| C13 | O | | | | | | | | | | | | | |
| C14 | | | | | | | | | | | | | | |

### 4.2.2. Development of Initial Reachability Matrix (IRM)

The SSIM table was converted to a binary matrix by substituting 1 when the two factors are related and 0 when they are not related, as shown in Table 6.

**Table 6.** Initial Reachability Matrix (IRM).

| CSFs | C1 | C2 | C3 | C4 | C5 | C6 | C7 | C8 | C9 | C10 | C11 | C12 | C13 | C14 |
|------|----|----|----|----|----|----|----|----|----|-----|-----|-----|-----|-----|
| C1 | 1 | 1 | 0 | 0 | 0 | 0 | 0 | 0 | 0 | 0 | 0 | 0 | 0 | 0 |
| C2 | 0 | 1 | 0 | 0 | 0 | 0 | 0 | 0 | 0 | 0 | 0 | 0 | 0 | 0 |
| C3 | 1 | 1 | 1 | 0 | 0 | 0 | 0 | 0 | 0 | 1 | 0 | 0 | 0 | 0 |
| C4 | 1 | 0 | 1 | 1 | 1 | 0 | 1 | 1 | 1 | 0 | 0 | 0 | 0 | 0 |
| C5 | 0 | 0 | 1 | 0 | 1 | 0 | 0 | 0 | 0 | 0 | 0 | 0 | 0 | 0 |
| C6 | 0 | 1 | 0 | 0 | 0 | 1 | 1 | 0 | 0 | 1 | 0 | 0 | 0 | 0 |
| C7 | 0 | 1 | 0 | 0 | 0 | 1 | 1 | 0 | 0 | 1 | 0 | 0 | 0 | 0 |
| C8 | 0 | 1 | 0 | 0 | 0 | 0 | 0 | 1 | 0 | 0 | 0 | 0 | 0 | 0 |
| C9 | 1 | 0 | 1 | 0 | 0 | 0 | 0 | 0 | 1 | 0 | 1 | 1 | 1 | 1 |
| C10 | 0 | 0 | 0 | 0 | 0 | 1 | 0 | 1 | 0 | 1 | 0 | 0 | 0 | 0 |
| C11 | 0 | 0 | 0 | 1 | 0 | 0 | 0 | 0 | 0 | 0 | 1 | 0 | 0 | 0 |
| C12 | 0 | 0 | 0 | 1 | 0 | 0 | 0 | 0 | 1 | 0 | 1 | 1 | 1 | 0 |
| C13 | 0 | 0 | 1 | 0 | 0 | 0 | 0 | 0 | 0 | 0 | 0 | 0 | 1 | 0 |
| C14 | 1 | 0 | 0 | 1 | 0 | 0 | 0 | 0 | 1 | 0 | 0 | 1 | 0 | 1 |

### 4.2.3. Development of Final Reachability Matrix (FRM)

A transitivity relationship is one that connects one CSF to another CSF without being directly related. For example, if Ci leads to Cj and Cj leads to Ck, the transitivity property implies that Ci leads to Ck. Computer coding was used to find the transitivity relationship of CSFs because it helps extract a complete and accurate relationship. From IRM as shown in Table 6, the computed result of FRM is as shown in Table 7, where 1* denotes the transitivity relationship.

**Table 7.** Final Reachability Matrix (FRM).

| CSFs | C1 | C2 | C3 | C4 | C5 | C6 | C7 | C8 | C9 | C10 | C11 | C12 | C13 | C14 | Dri. |
|------|-----|-----|-----|-----|-----|-----|-----|-----|-----|------|------|------|------|------|------|
| C1 | 1 | 1 | 0 | 0 | 0 | 0 | 0 | 0 | 0 | 0 | 0 | 0 | 0 | 0 | 2 |
| C2 | 0 | 1 | 0 | 0 | 0 | 0 | 0 | 0 | 0 | 0 | 0 | 0 | 0 | 0 | 1 |
| C3 | 1 | 1 | 1 | 0 | 0 | 1* | 1* | 1* | 0 | 1 | 0 | 0 | 0 | 0 | 7 |
| C4 | 1 | 1* | 1 | 1 | 1 | 1* | 1 | 1 | 1 | 1* | 1* | 1* | 1* | 1* | 14 |
| C5 | 1* | 1* | 1 | 0 | 1 | 1* | 1* | 1* | 0 | 1* | 0 | 0 | 0 | 0 | 8 |
| C6 | 0 | 1 | 0 | 0 | 0 | 1 | 1 | 1* | 0 | 1 | 0 | 0 | 0 | 0 | 5 |
| C7 | 0 | 1 | 0 | 0 | 0 | 1 | 1 | 1* | 0 | 1 | 0 | 0 | 0 | 0 | 5 |
| C8 | 0 | 1 | 0 | 0 | 0 | 0 | 0 | 1 | 0 | 0 | 0 | 0 | 0 | 0 | 2 |
| C9 | 1 | 1* | 1 | 1* | 1* | 1* | 1* | 1* | 1 | 1* | 1 | 1 | 1 | 1 | 14 |
| C10 | 0 | 1* | 0 | 0 | 0 | 1 | 1* | 1 | 0 | 1 | 0 | 0 | 0 | 0 | 5 |
| C11 | 1* | 1* | 1* | 1 | 1* | 1* | 1* | 1* | 1* | 1* | 1 | 1* | 1* | 1* | 14 |
| C12 | 1* | 1* | 1* | 1 | 1* | 1* | 1* | 1* | 1 | 1* | 1 | 1 | 1 | 1* | 14 |
| C13 | 1* | 1* | 1 | 0 | 0 | 1* | 1* | 1* | 0 | 1* | 0 | 0 | 1 | 0 | 8 |
| C14 | 1 | 1* | 1* | 1 | 1* | 1* | 1* | 1* | 1 | 1* | 1* | 1 | 1* | 1 | 14 |
| Dep. | 9 | 14 | 8 | 5 | 6 | 11 | 11 | 12 | 5 | 11 | 5 | 5 | 6 | 5 | |

1* denotes the transitivity relationship. Dri. and Dep. are the driving and dependence powers, respectively.

### 4.2.4. Process of the FRM to Level Partitions

From Table 7, the final reachability matrix was assessed to determine the level partition of 14 CSFs. Table 8 shows the level partition, which consists of the reachability and antecedent sets. The reachability set includes direct and transitivity relationships in the row, and the antecedent set represents direct and transitivity relationships as defined in the column. The intersection set is the group of members that belong to both the reachability and the antecedent set. Any CFS with an intersection set equal to the reachability set will be removed from the level partition. Then, the canonical matrix is conducted by including the CFSs that have been removed from the level partition. The processes were repeated until all CSFs were deducted from the level partition. If CSFs were left, the above procedure was invalid [38,56]. This work was repeated six times until reaching TISM at level 6. The results implied that 14 CSFs were considered for the traceability operation, which was divided into six levels, as summarized in Table 9.

**Table 8.** Level partition on reachability matrix.

| CSFs | Reachability Set | Antecedent Set | Intersection | Level |
|------|------------------|----------------|--------------|-------|
| C1 | 1,2 | 1,3,4,5,9,11,12,13,14 | | |
| C2 | 2 | 1,2,3,4,5,6,7,8,9,10,11,12,13,14 | 2 | I |
| C3 | 1,2,3,6,7,8,10 | 3,4,5,9,11,12,13,14 | | |
| C4 | 1,2,3,4,5,6,7,8,9,10,11,12,13,14 | 4,9,11,12,14 | | |
| C5 | 1,2,3,5,6,7,8,10 | 4,5,9,11,12,14 | | |
| C6 | 2,6,7,8,10 | 3,4,5,6,7,9,10,11,12,13,14 | | |
| C7 | 2,6,7,8,10 | 3,4,5,6,7,9,10,11,12,13,14 | | |
| C8 | 2,8 | 3,4,5,6,7,8,9,10,11,12,13,14 | | |
| C9 | 1,2,3,4,5,6,7,8,9,10,11,12,13,14 | 4,9,11,12,14 | | |
| C10 | 2,6,7,8,10 | 3,4,5,6,7,9,10,11,12,13,14 | | |
| C11 | 1,2,3,4,5,6,7,8,9,10,11,12,13,14 | 4,9,11,12,14 | | |
| C12 | 1,2,3,4,5,6,7,8,9,10,11,12,13,14 | 4,9,11,12,14 | | |
| C13 | 1,2,3,6,7,8,10,13 | 4,9,11,12,13,14 | | |
| C14 | 1,2,3,4,5,6,7,8,9,10,11,12,13,14 | 4,9,11,12,14 | | |

**Table 9.** Canonical matrix.

| CSFs | C2 | C1 | C8 | C6 | C7 | C10 | C3 | C5 | C13 | C4 | C9 | C11 | C12 | C14 | Level |
|------|----|----|----|----|----|-----|----|----|-----|----|----|-----|-----|-----|-------|
| C2 | 1 | 0 | 0 | 0 | 0 | 0 | 0 | 0 | 0 | 0 | 0 | 0 | 0 | 0 | I |
| C1 | 1 | 1 | 0 | 0 | 0 | 0 | 0 | 0 | 0 | 0 | 0 | 0 | 0 | 0 | II |
| C8 | 1 | 0 | 1 | 0 | 0 | 0 | 0 | 0 | 0 | 0 | 0 | 0 | 0 | 0 | II |
| C6 | 1 | 0 | 1* | 1 | 1 | 1 | 0 | 0 | 0 | 0 | 0 | 0 | 0 | 0 | III |
| C7 | 1 | 0 | 1* | 1 | 1 | 1 | 0 | 0 | 0 | 0 | 0 | 0 | 0 | 0 | III |
| C10 | 1* | 0 | 1 | 1 | 1* | 1 | 0 | 0 | 0 | 0 | 0 | 0 | 0 | 0 | III |
| C3 | 1 | 1 | 1* | 1* | 1* | 1 | 1 | 0 | 0 | 0 | 0 | 0 | 0 | 0 | IV |
| C5 | 1* | 1* | 1* | 1* | 1* | 1* | 1 | 1 | 0 | 0 | 0 | 0 | 0 | 0 | V |
| C13 | 1* | 1* | 1* | 1* | 1* | 1* | 1 | 0 | 1 | 0 | 0 | 0 | 0 | 0 | V |
| C4 | 1* | 1 | 1 | 1* | 1 | 1* | 1 | 1 | 1* | 1 | 1 | 1* | 1* | 1* | VI |
| C9 | 1* | 1 | 1* | 1* | 1* | 1* | 1 | 1* | 1 | 1* | 1 | 1 | 1 | 1 | VI |
| C11 | 1* | 1* | 1* | 1* | 1* | 1* | 1* | 1* | 1* | 1 | 1* | 1 | 1* | 1* | VI |
| C12 | 1* | 1* | 1* | 1* | 1* | 1* | 1* | 1* | 1 | 1 | 1 | 1 | 1 | 1* | VI |
| C14 | 1* | 1 | 1* | 1* | 1* | 1* | 1* | 1* | 1* | 1 | 1 | 1* | 1 | 1 | VI |
| Level | I | II | II | III | III | III | IV | V | V | VI | VI | VI | VI | VI | |

1* denotes the transitivity relationship.

Factor clustering in the canonical matrix helps group the factors at each level and shows the relationship for each factor from the FRM. Furthermore, it also helps verify and make a complete TISM diagram of the relationship, as shown in Table 9.

4.2.5. Development of Total Interpretive Structural Modeling (TISM)

TISM was developed from ISM. TISM is designed to explain and interpret the relationship between CSFs to reduce ambiguity [54] in supply chain and technology research [38,56–59]. TISM used questions (what, why, and how) to understand how the factors interact in the traceability system [55,60]. The canonical matrix supports creating a TISM model that divides the group of CSFs into a hierarchy of traceability implementations. A relationship line explains the reasons for such a relationship in each element. Figure 3 shows that the TISM model has six levels, and the System Reliability factor is at the top of the model. It can imply that the most important aspect of a traceability system is System Reliability [36]. The color of TISM simplifies the operations and assignments of the organization in three contexts based on the TOE framework.

Between the related CSFs, there is an arrow line showing the relationship between the CSFs and explaining why they are related.

*4.3. Fuzzy MICMAC Analysis*

4.3.1. Development of the Binary Direct Relationship Matrix

Fuzzy MICMAC must be prepared for analysis by the binary direct relationship matrix (BDRM) process. The FRM table is constructed by overlooking the transitivity relationship and transforming all diagonal entries from 1 to 0 [56].

4.3.2. Process of the Fuzzy Direct Relationship Matrix (FDRM)

The level of relationship described by the MICMAC method is still an issue, and the level of relationship should be considered. The traditional correlation between the CSFs gives a binary score of 1 and 0, which means that the two variables are influenced or uninfluenced, respectively. The fuzzy MICMAC was developed using the linguistic scale principle, which increases the degree of correlation in a relationship equal to 1. Experts rated the correlation level based on the variable, which represented a degree of membership weight for each pair of relationships by using the linguistic scale principle shown in Table 10 [60]. The results of the rating are shown in Table 11. The scale was calculated using a qualitative method by defuzzification, which converted a crisp number from the fuzzy set [58,61].

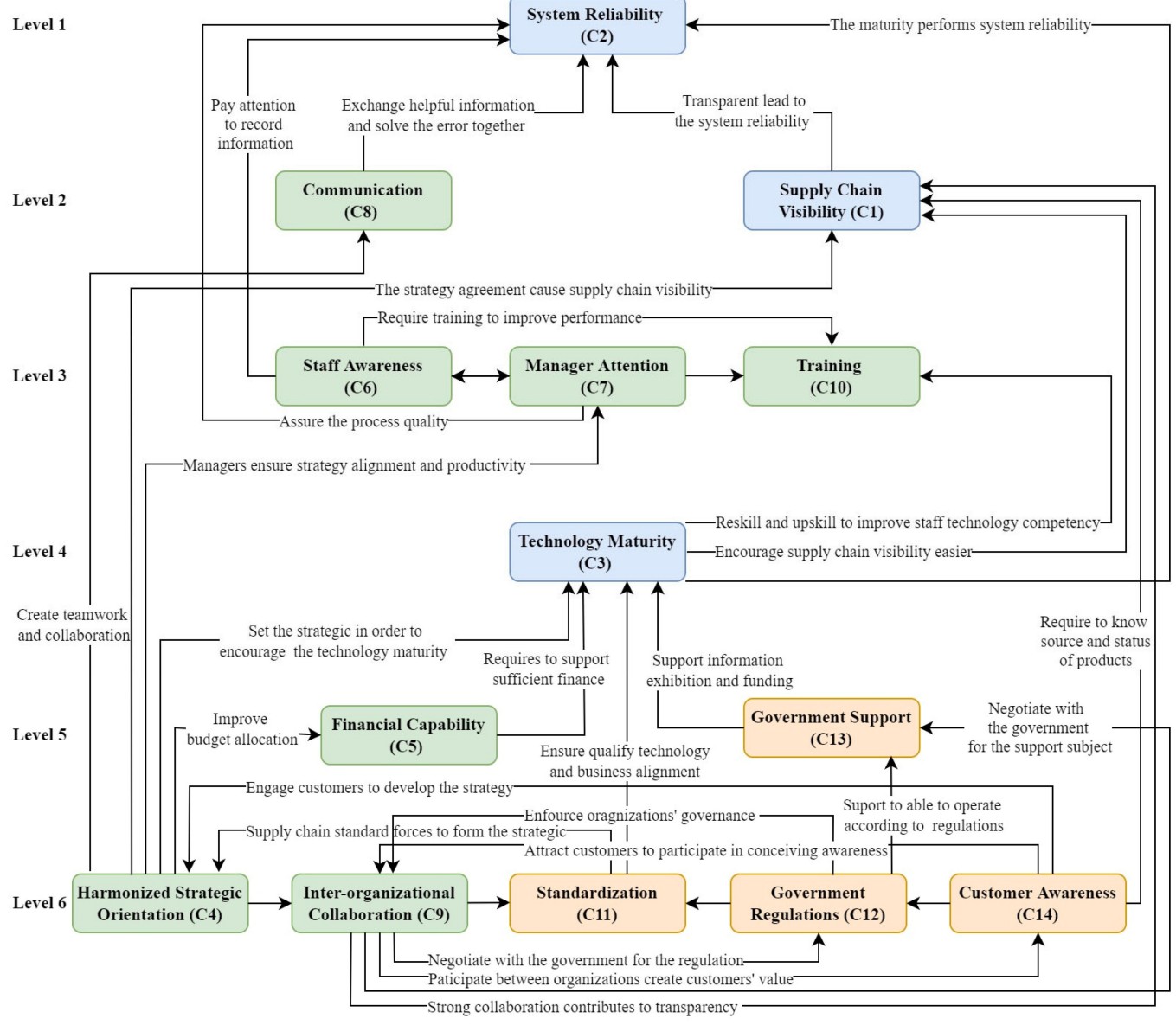

**Figure 3.** TISM for innovation framework of MOCT.

**Table 10.** Linguistic scale for fuzzy MICMAC analysis.

| Relationship | No Influence | Very Week | Week | Medium | Strong | Very Strong | Complete Influence |
|---|---|---|---|---|---|---|---|
| Scale value | 0 | 0.1 | 0.3 | 0.5 | 0.7 | 0.9 | 1 |

### 4.3.3. Development of Fuzzy Stabilized Matrix

After the degree of correlation between CSFs was acquired, the direct and transitive relationship effect was assessed using fuzzy MICMAC relationships. The FDRM was multiplied for a stabilized matrix with (4) until the weights of dependence and driving powers were stabilized [38,58]. The results are shown in Table 12. The driving power and dependence power in the fuzzy stabilize matrix were derived by summing up the values in each row and column, respectively.

**Table 11.** FDRM for CFSs influencing traceability system.

| CSFs | C1 | C2 | C3 | C4 | C5 | C6 | C7 | C8 | C9 | C10 | C11 | C12 | C13 | C14 |
|---|---|---|---|---|---|---|---|---|---|---|---|---|---|---|
| C1 | 0 | 0.9 | 0 | 0 | 0 | 0 | 0 | 0 | 0 | 0 | 0 | 0 | 0 | 0 |
| C2 | 0 | 0 | 0 | 0 | 0 | 0 | 0 | 0 | 0 | 0 | 0 | 0 | 0 | 0 |
| C3 | 0.7 | 0.9 | 0 | 0 | 0 | 0 | 0 | 0 | 0 | 0.7 | 0 | 0 | 0 | 0 |
| C4 | 0.7 | 0 | 0.5 | 0 | 0.7 | 0 | 0.7 | 0.7 | 0.7 | 0 | 0 | 0 | 0 | 0 |
| C5 | 0 | 0 | 0.6 | 0 | 0 | 0 | 0 | 0 | 0 | 0 | 0 | 0 | 0 | 0 |
| C6 | 0 | 0.7 | 0 | 0 | 0 | 0 | 0.7 | 0 | 0 | 0.7 | 0 | 0 | 0 | 0 |
| C7 | 0 | 0.7 | 0 | 0 | 0 | 0.7 | 0 | 0 | 0 | 0.7 | 0 | 0 | 0 | 0 |
| C8 | 0 | 0.5 | 0 | 0 | 0 | 0 | 0 | 0 | 0 | 0 | 0 | 0 | 0 | 0 |
| C9 | 0.7 | 0 | 0.5 | 0 | 0 | 0 | 0 | 0 | 0 | 0 | 0.7 | 0.5 | 0.5 | 0.3 |
| C10 | 0 | 0 | 0 | 0 | 0 | 0.7 | 0 | 0.7 | 0 | 0 | 0 | 0 | 0 | 0 |
| C11 | 0 | 0 | 0 | 0.9 | 0 | 0 | 0 | 0 | 0 | 0 | 0 | 0 | 0 | 0 |
| C12 | 0 | 0 | 0 | 0.6 | 0 | 0 | 0 | 0 | 0.7 | 0 | 0.7 | 0 | 0.9 | 0 |
| C13 | 0 | 0 | 0.3 | 0 | 0 | 0 | 0 | 0 | 0 | 0 | 0 | 0 | 0 | 0 |
| C14 | 0.7 | 0 | 0 | 0.9 | 0 | 0 | 0 | 0 | 0.7 | 0 | 0 | 0.3 | 0 | 0 |

**Table 12.** Fuzzy stabilized matrix.

| CSFs | C1 | C2 | C3 | C4 | C5 | C6 | C7 | C8 | C9 | C10 | C11 | C12 | C13 | C14 | Dri. |
|---|---|---|---|---|---|---|---|---|---|---|---|---|---|---|---|
| C1 | 0 | 0.9 | 0 | 0 | 0 | 0 | 0 | 0 | 0 | 0 | 0 | 0 | 0 | 0 | 0.9 |
| C2 | 0 | 0 | 0 | 0 | 0 | 0 | 0 | 0 | 0 | 0 | 0 | 0 | 0 | 0 | 0 |
| C3 | 0.7 | 0.9 | 0 | 0 | 0 | 0.7 | 0.7 | 0.7 | 0 | 0.7 | 0 | 0 | 0 | 0 | 4.4 |
| C4 | 0.7 | 0.7 | 0.6 | 0.7 | 0.7 | 0.7 | 0.7 | 0.7 | 0.7 | 0.7 | 0.7 | 0.5 | 0.5 | 0.3 | 8.9 |
| C5 | 0.6 | 0.6 | 0.6 | 0 | 0 | 0.6 | 0.6 | 0.6 | 0 | 0.6 | 0 | 0 | 0 | 0 | 4.2 |
| C6 | 0 | 0.7 | 0 | 0 | 0 | 0.7 | 0.7 | 0.7 | 0 | 0.7 | 0 | 0 | 0 | 0 | 3.5 |
| C7 | 0 | 0.7 | 0 | 0 | 0 | 0.7 | 0.7 | 0.7 | 0 | 0.7 | 0 | 0 | 0 | 0 | 3.5 |
| C8 | 0 | 0.5 | 0 | 0 | 0 | 0 | 0 | 0 | 0 | 0 | 0 | 0 | 0 | 0 | 0.5 |
| C9 | 0.7 | 0.7 | 0.6 | 0.7 | 0.7 | 0.7 | 0.7 | 0.7 | 0.7 | 0.7 | 0.7 | 0.5 | 0.5 | 0.3 | 8.9 |
| C10 | 0 | 0.7 | 0 | 0 | 0 | 0.7 | 0.7 | 0.7 | 0 | 0.7 | 0 | 0 | 0 | 0 | 3.5 |
| C11 | 0.7 | 0.7 | 0.6 | 0.9 | 0.7 | 0.7 | 0.7 | 0.7 | 0.7 | 0.7 | 0.7 | 0.5 | 0.5 | 0.3 | 9.1 |
| C12 | 0.7 | 0.7 | 0.6 | 0.7 | 0.7 | 0.7 | 0.7 | 0.7 | 0.7 | 0.7 | 0.7 | 0.5 | 0.9 | 0.3 | 9.3 |
| C13 | 0.3 | 0.3 | 0.3 | 0 | 0 | 0.3 | 0.3 | 0.3 | 0 | 0.3 | 0 | 0 | 0 | 0 | 2.1 |
| C14 | 0.7 | 0.7 | 0.6 | 0.9 | 0.7 | 0.7 | 0.7 | 0.7 | 0.7 | 0.7 | 0.7 | 0.5 | 0.5 | 0.3 | 9.1 |
| Dep. | 5.1 | 8.8 | 3.9 | 3.9 | 3.5 | 7.2 | 7.2 | 7.2 | 3.5 | 7.2 | 3.5 | 2.5 | 2.9 | 1.5 | |

The formula used to calculate the strength of the indirect relationships is as follows:

$$m_c(i,j) = \max_{k=1}^{n}[\min\{m_1(i,k), m_{c-1}(k,j)\}],$$
$$i,j = 1, 2, \ldots, n; \ c = 2, 3, \ldots, \tag{4}$$

where $m_c(i,k)$ represents the direct relationship strength between parameters i and k.

4.3.4. Creation of the Conclusive Dependence–Driving Diagram

The fuzzy MICMAC process analyzed the relationship between CSFs using a linguistic scale to obtain the dependence–driving power diagram. After obtaining the diagram, the power was plotted in a conclusive diagram that separated CSFs into four clusters, comprising autonomous causes, dependent causes, linkage causes, and independent causes, as shown in Figure 4.

Cluster I, consisting of autonomous causes, was characterized by CSFs with low dependence and driver power. This cluster has three CSFs: C3, C5, and C13.

Cluster II, including independent causes, was characterized by CSFs with high driving power but low dependence power. Five CSFs were grouped in this cluster: C4, C11, C9, C12, and C14.

Cluster III, which had linkage causes, was distinguished by CSFs with high dependence and driving power. Therefore, no CSFs were assigned in this cluster due to no power.

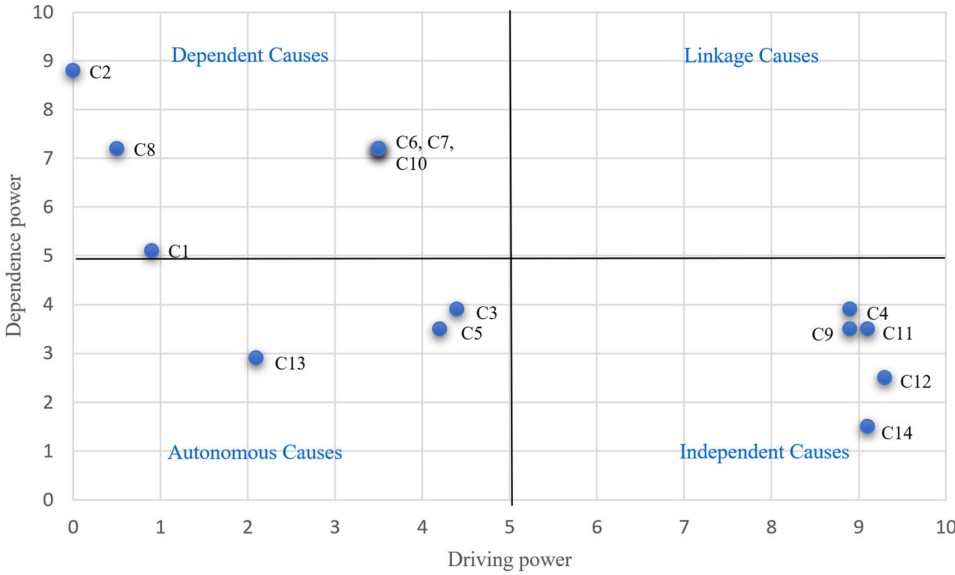

**Figure 4.** Conclusive dependence–driving power diagram.

Cluster IV, having a dependent cause, was characterized by CSFs with high dependence but low driving power. This cluster had six CSFs: C2, C8, C1, C6, C7, and C10.

## 5. Discussion

An in-depth discussion of the proposed framework is provided in this section, which consists of three subsections. TISM of the proposed framework is described and discussed in the first subsection. The conclusive dependence–driving power diagram of the CFFs is discussed in the second subsection. Then, a combined discussion of TISM and the conclusive dependence–driving power diagram is presented in the last subsection. Details of the in-depth discussion are provided as follows.

### 5.1. TISM for Innovation Framework of Medical Organic Cannabis Traceability

As shown in Figure 3, the TISM of the proposed framework represents the relationship between the CSFs. We can see that the 14 CSFs were constructed to be a six-level hierarchy framework. To describe the characteristics of TISM, it is necessary to start from the bottom of the hierarchy.

The bottom level, Level 6, is the basement of the framework. It consists of five factors, including Harmonized Strategic Orientation (C4), Standardization (C9), Inter-organizational Collaboration (C11), Government Regulations (C12), and Customer Awareness (14). The relationships between factors are represented by a solid line with an arrow called "relationship line". Note that only direct relationships are shown in this figure, whereas indirect relationships are neglected to prevent confusion. The reason for each relationship is presented as the statement given on the relationship line. For example, Harmonized Strategic Orientation has a direct relationship with Standardization because the related standards drive the strategy of the associated organization. The CSFs at this level have relationships with each other and also on the upper level.

The upper level is Level 5. It consists of two factors, Financial Capability (C5) and Government Support (C13). These factors are driven by the factors in Level 6 and influence the upper level, which is Level 4 in this case. For Level 4, there is only one factor, Technology Maturity (C3). It can imply that this factor provides a connection between the lower level (Levels 5–6) and the higher level (Levels 1–3). This factor must be achieved before we can

accomplish the upper-level factors. In Level 3, there are three factors: Staff Awareness (C6), Manager Attention (C7), and Training (C10). We can see that this level involves human resources in the related organization. These factors are the basement of the upper level. Level 2 consists of two factors, Communication (C8) and Supply Chain Visibility (C1). Supply Chain Visibility is driven by Technology Maturity, Harmonized Strategic Orientation, Standardization, and Customer Awareness, while Communication is driven only by Harmonized Strategic Orientation.

The top level of this proposed framework is Level 1. Only one factor has been placed at this level, which is System Reliability (C2). We can imply that this factor is affected by all lower lever factors in an either direct or indirect way. System Reliability is the most important factor for MOCT. It is a key factor for the innovation framework because it represents an indicator of the efficiency and quality of people's awareness and technological devices [62,63].

Moreover, in this framework, each factor is labeled with a color according to the TOE theory, where blue, green, and orange represent technological factors, organizational factors, and environmental factors, respectively. Therefore, we can trace back how TOE has an effect on the proposed framework.

When organizations want to implement the MOCT system, they have to start from the bottom level, as described previously. The Harmonized Strategic Orientation and Inter-organizational Collaboration must be considered so that stakeholders have the same vision across the supply chain. It is necessary to take time and pay attention to these factors because they involve all related stakeholders, generally from different organizations. Traceability standards must be considered and agreed upon between stakeholders. Government rules and customer requirements are also crucial for this consideration. The company's financial capabilities and government support, for example, knowledge and funding, will be a factor in supporting organizations' decision to adopt mature technology. On the part of the organization, Staff Awareness, Manager Attention, Communication, and Training will encourage visibility in the supply chain, which will lead to a reliable traceability system [64]. Each organization must support the traceability system and make it easy to use by eliminating redundant processes. This innovation framework will help governments control how medical cannabis is used and ensure its safety.

*5.2. Conclusive Dependence–Driving Power Diagram*

The dependence–driving power diagram is a diagram that explains the power of CSFs. In this study, factors appear in three clusters, as shown in Figure 4. Cluster I is the autonomous cause. These factors are independent of the other factors and have little influence on them. However, Government Support (C13) has lower driving and dependence power than others. Thus, Technology Maturity (C3) and Financial Capability (C5) may have a significant impact on the system's success. Cluster II is an independent cause with high driving power but low dependent power. It means that the factors, i.e., Harmonized Strategic Orientation (C4), Standardization (C11), Inter-organizational Collaboration (C9), Government Regulations (C12), and Customer Awareness (C14), will affect or push other factors to complete the traceability system [38]. No CSF falls under Cluster III (linkage cause). This means that none of the CSFs can work on their own. Each factor can affect the other factors, and they also impact themselves. Consequently, they require cautious management. Cluster IV, which is the dependent cause cluster, consists of System Reliability (C2), Communication (C8), Supply Chain Visibility (C1), Staff Awareness (C6), Manager Attention (C7), and Training (C10). The CSFs have low driving power but high dependent power, which is controlled by factors such as Government Regulations, Standardization, and Customer Awareness.

The diagram indicates that Government Regulation has the most influence. This factor must be implemented proactively through the regulations that drive the MOCT system. In addition, Customer Awareness also has high driving power and the lowest dependence power. This factor is related to many customers, especially in the medical domain, who

want to use cannabis free of heavy metals. On the other hand, System Reliability is a factor that has no driving power and relies on other factors. Finally, Staff Awareness, Manager Attention, and Training factors, which are organizational variables, have the same driving and dependence powers. They can be achieved through other factors and must be combined to be effective.

### 5.3. Comparison of TISM and Conclusive Dependence–Driving Power Diagram

From the results of the TISM and conclusive diagram, we can see that these results agree regarding CSF grouping and the relationship between the CSFs. The agreement between TISM and the conclusive diagram can be demonstrated as follows.

Firstly, all CSFs at Level 6 of TISM appear in the cluster of independent causes with high driving power but low dependent power. These CSFs have a high driving power for supply chain performance, which means that they can push the other CSFs to success in implementing traceability. The findings are consistent with previous research on food traceability, which found that government regulation plays a role at the first level of action and that the customer factor appears as an independent cause [38]. Moreover, traceability success depends on legal and customer factors [40]. Inter-organizational Collaboration challenges the ability to persuade and exchange data among the stakeholders, including demand, supply, and knowledge, which can be advantageous in the medical cannabis business [65].

The CSFs located in Levels 1–3 of TISM are the members of the dependent causes of the conclusive dependence–driving power diagram. These CSFs have strong dependence power on the other CSFs but weak driving power. This means that they have to rely on other CSFs to succeed in traceability implementation.

Lastly, Government Support, Technology Maturity, and Financial Capability are lightly dependent on each other. These CSFs have a moderate effect on the success of the traceability process. Therefore, we are confident that the analysis can be effectively used in planning the implementation of traceability systems.

### 5.4. TISM and Its Relation with Open Innovation

TISM has become the most widely accepted method today. It has been used across various disciplines to show the relationship between CSFs, as illustrated following. In industrial research, this method can identify and prioritize the CSFs that influence the success of the oil and automotive supply chain industries [66,67]. Furthermore, it can define not only the success factors but also the barriers to the success of Industry 4.0 [68,69]. CSF classification can be done using this method, such as [70–72] for intelligent manufacturing systems and environmentally friendly innovations in the manufacturing industry, including remanufacturing. TISM has been used in health-care research to identify CSFs for clinical decision support systems by machine learning and health-care system development [73–75]. It was implemented in logistics and supply chains to create a framework for the joint development supply chain, increasing the competitive advantage [76–78]. The social aspect used this approach to develop a model for sustainable investment and home planning [79,80]. Environmental research adopted this method to identify solutions to reduce energy consumption and pollution [81,82]. Nevertheless, this approach was applied to knowledge management disciplines, human resource management, and technology-related risk management [83–86], including analysis of factors affecting the effectiveness of e-learning [87].

In summary, this work is based on TISM. The research finding can be used as a guideline for developing countries such as Thailand to ensure that the medical cannabis industry is reliable and safe. This framework will allow the medical cannabis supply chain to have a visible product quality standard where customers can trace product quality data from upstream to downstream. Stakeholders' belief in medicinal cannabis products will ensure buying and selling in a sustainable supply chain. Moreover, the framework is a mechanism that helps search for the appropriate suppliers in the supply chain to improve

the traceability system. It allows managers of all supply chain stakeholders who have a stake in the traceability system to raise awareness and plan for its implementation.

This work can be useful in many fields in the future. For example, from a social perspective, supply chain stakeholders implementing a traceability system can study this framework's sequential implementation to plan their sustainability. Furthermore, from an education perspective, the framework can be validated using structural equation modeling (SEM) and can be used as an initial study for further work. In countries with different contexts than this research, there may be different factors affecting the success of MOCT. In addition, research may examine strategies and operations for intrinsic factors such as the synchronized strategic orientation of supply chains.

The limitations of this study are that medical cannabis has only been recognized in the past few years and only in a few countries. However, previous research has already been reviewed to identify the underlying factors. Furthermore, in determining the relationship between CSFs, data cannot be collected from the general public. Therefore, purposive sampling was used to screen experienced experts who are involved in medical cannabis to consider CSF relationships to create the innovation framework.

## 6. Conclusions

This work identified the relationship between the CSFs of medical cannabis traceability considering organic-grade cannabis in the DSC. Structural modeling was conducted through qualitative and quantitative methods. According to the research questions of this work, the findings are as follows. Firstly, this study could identify the CSFs of MOCT in the supply chain by using the literature and could verify them with medical cannabis experts covering the perspectives of government regulators, policymakers, business operators, and customers. To reduce redundancy factors and increase contextual content validity, multi-rater kappa statistics were used until fourteen CSFs were accepted. Secondly, this study conducted an innovation framework of CSFs of MOCT in the DSC with TISM. Fourteen CSFs were analyzed and categorized into six hierarchical groups of actions. The relationship line describes the connection of each CSF. It was found that the System Reliability factor was the ultimate goal of implementation for the organic medical cannabis traceability system. The CSFs that should be considered in the first action of implementation were Harmonized Strategic Orientation, Inter-organizational Collaboration, Standardization, Government Regulations, and Customer Awareness. Lastly, this work classified factors based on their driving and dependence powers with fuzzy MICMAC from experts' opinions for elaborating the dependence–driving power. The CSFs on the conclusive dependence–driving power diagram appeared in three clusters: autonomous, dependent, and independent. In conclusion, the CSFs have consistent relationships and integrity and are appropriate for further development.

**Author Contributions:** Conceptualization, W.P. and T.S.; methodology, W.P. and T.S.; software, W.P. and T.S.; validation, W.P.; formal analysis, W.P. and T.S.; resources, W.P. and T.S.; data curation, W.P.; writing—original draft preparation, W.P.; writing—review and editing, W.P. and T.S.; visualization, W.P. and T.S.; supervision, T.S.; project administration, W.P. and T.S. All authors have read and agreed to the published version of the manuscript.

**Funding:** This work was supported in parts through a Ph.D. scholarship from the Ministry of Higher Education, Science, Research, and Innovation (MHESI) of the Royal Thai Government.

**Institutional Review Board Statement:** The study was conducted in accordance with the Declaration of Helsinki and approved by the Ethics Committee of Mahidol University (protocol code 2021/403.1309, 27 September 2021).

**Informed Consent Statement:** Informed consent was obtained from all subjects involved in the study.

**Data Availability Statement:** The data presented in this study are available on request from the corresponding author. The data are not publicly available due to privacy and ethical concerns; neither the data nor the source of the data can be made available under Mahidol University.

**Acknowledgments:** We would like to thank anonymous experts from the private and public sectors for their opinions on the data analysis. We would like to thank Supaporn Kiattisin and Theeraya Mayakol for their suggestion and recommendation. This work was supported in parts through a Ph.D. scholarship from the Ministry of Higher Education, Science, Research, and Innovation (MHESI) of the Royal Thai Government.

**Conflicts of Interest:** The authors declare no conflict of interest.

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
