# Peer review of "An Innovation Framework of Medical Organic Cannabis Traceability in Digital Supply Chain"

_2199-8531, doi:10.3390/joitmc8040196_

Round 1

Reviewer 1 Report

Thank you for the opportunity to comment this paper entitled: " An Innovation Framework of Medical Organic Cannabis Traceability in Digital Supply Chain". I found the empirical setting very interesting as it raised multiple implications from a social, technological and regulatory point of view. Moreover, it is a current issue considering that many countries are still debating about the legalization of cannabis for medical purposes. The critical success factors identified and their rationalization in the TSIM framework could improve the knowledge in the field and provide suggestion to policy makers and managers.

However, I would suggest some improvement mainly in the literature review and discussion. Please find below my comments.

-          Literature review is a fundamental part of the paper since critical success factors derive from it. For this reason, claiming to have done a structured literature review is not sufficient. My suggestion is to insert a section in which you will describe in detail how the papers were selected and how they are analyzed to identify the critical success factors. In particular, if a keyword search was carried out, what are the keywords chosen, on which databases, with what limits applied to the search, how were the identified papers then selected? Once the papers have been selected, how were them analysed and the critical success factors extracted from them?

-          In the discussion I would like the “TSIM for innovation framework” to be better explained. The theoretical model developed seems very rich in information but too little space is dedicated to its explanation and deepening. My suggestion is to broaden the discussion of the model to better explain it to the reader and derive any theoretical and practical contributions from it.

-          Similarly, the “Conclusive dependence-driving power diagram” is only marginally explained. Again I suggest broadening the discussion as suggested for the TSIM.

In conclusion, I find the paper interesting but the way the literature review was conducted is ambiguous and the discussion underdeveloped. I suggest to revise and resubmit.

Author Response

Dear Reviewer 

Thanks for your valuable comments.  We did make corrections carefully.  Please check the attached file.

Best Regards.

Authors 

Reviewer 2 Report

1. Introduction: must clearly state the contribution of the research; the author may identify the contribution of the article in a separate subsection.

2. Before mentioning that "this session has sub session 2.1,2.2....," the author of must specify the review's organizational structure. An overview of the sections to be  either in the form of a diagram or a brief explanation.

3.The academic and professional/social implications of your research must be made crystal obvious.

Author Response

(The authors gave the same response as above.)

Round 2

Reviewer 1 Report

All modifications have been properly addressed and I have no further comments.

Author Response

Dear Reviewer

English has been checked by AUTHORServices of MPDI.  Please see the attached file

Regards

Reviewer 2 Report

Well-done 

Author Response

(The authors gave the same response as above.)
